# Immunogenicity and Safety of SARS-CoV-2 Protein Subunit Recombinant Vaccine (IndoVac^®^) as a Heterologous Booster Dose against COVID-19 in Indonesian Adolescents

**DOI:** 10.3390/vaccines12080938

**Published:** 2024-08-22

**Authors:** Eddy Fadlyana, Kusnandi Rusmil, Muhammad Gilang Dwi Putra, Frizka Primadewi Fulendry, Nitta Kurniati Somantri, Alvira Dwilestarie Putri, Rini Mulia Sari, Mita Puspita, Gianita Puspita Dewi

**Affiliations:** 1Clinical Research Unit, Growth and Development–Social Pediatrics Division, Department of Child Health, Faculty of Medicine, Universitas Padjadjaran, Hasan Sadikin Hospital, Bandung 40161, Indonesia; kusnandi.rusmil2017@unpad.ac.id (K.R.); gilangdwiputra@gmail.com (M.G.D.P.); frizkaapf@gmail.com (F.P.F.); alvira16001@mail.unpad.ac.id (A.D.P.); 2Garuda Primary Health Care Centres, Bandung City Health Service, Bandung 40161, Indonesia; nitta.bartok@yahoo.com; 3Surveillance and Clinical Trial Division, PT Bio Farma, Bandung 40161, Indonesia; rini.mulia@biofarma.co.id (R.M.S.); mita.puspita@biofarma.co.id (M.P.); gianita.dewi@biofarma.co.id (G.P.D.)

**Keywords:** adolescent, booster, COVID-19 vaccine, IndoVac^®^, protein recombinant

## Abstract

Adolescents are vulnerable to Coronavirus disease 2019 (COVID-19) infections; thus, their antibodies should be maintained above the protective value. This study aimed to evaluate the immune response and safety to the SARS-CoV-2 protein subunit recombinant vaccine (IndoVac^®^) as a heterologous booster dose against COVID-19 in Indonesian adolescents. This open-label prospective intervention study enrolled 150 clinically healthy adolescents aged 12–17 years who had received complete primary doses of the CoronaVac^®^ vaccine from Garuda Primary Care Centres in Bandung City. The result of immunogenicity was presented with a 95% confidence interval (CI) and analyzed with *t*-tests from 14 days and 3, 6, and 12 months. The neutralizing antibody geometric mean titers (GMTs) (IU/mL) at baseline and 14 days after booster dose were 303.26 and 2661.2, respectively. The geometric mean fold rises (GMFR) at 3, 6, and 12 months after booster dose were 6.67 (5.217–8.536), 3.87 (3.068–4.886), and 2.87 (2.232–3.685), respectively. Both the neutralizing antibody and IgG antibody were markedly higher in the adolescents than in the adults at every timepoint. The incidence rate of adverse effects (AEs) until 28 days after booster dose was 82.7%, with a higher number of local events reported. Most reported solicited AEs were local pain followed by myalgia with mild intensity. Unsolicited AEs varied with each of the incidence rates < 10%, mostly with mild intensity. Adverse events of special interest (AESI) were not observed. At the 12-month follow-up after the booster dose, four serious adverse events (SAEs) not related to investigational products and research procedures were noted. This study showed that IndoVac^®^ has a favorable immunogenicity and safety profile as a booster in adolescents and that the antibody titer decreases over time.

## 1. Introduction

Coronavirus disease 2019 (COVID-19) is a respiratory infectious disease caused by severe acute respiratory syndrome coronavirus 2 (SARS-CoV-2) that has become a global burden [1,2]. The World Health Organization (WHO) has reported 775,364,261 confirmed COVID-19 cases and 7,046,320 deaths worldwide as of 21 April 2024 [1,3]. A previous study showed that in 2020, there were 37,705 reported confirmed COVID-19 cases among children in Indonesia, of which 175 resulted in death, with the highest mortality from children aged 10–18 years [4]. Moreover, as of 21 December 2020, the Indonesian Pediatric Society has reported 35,506 suspected cases of COVID-19 in children, of which 522 resulted in death (case fatality ratio: 1.4) [4]. The prevalence of COVID-19 is significantly lower in children than in adults; however, pediatric disease is more probably underdiagnosed because of the high number of asymptomatic or mild cases [5,6]. Additionally, children are vulnerable to family cluster outbreaks and can be a source of transmission of SARS-CoV-2 to each other and adult family members [7,8]. A small proportion of children require hospitalization or intensive care, and pediatric cases of COVID-19-associated multisystem inflammatory syndrome (MIS-C) and long COVID-19 have been reported [9,10]. Moreover, COVID-19 has affected the behavior and psychological aspects of adolescents, especially during the lockdown period, and social isolation, which may have led to symptoms of anxiety, feelings of loneliness, irritability, boredom, fear, stress, anger, and depression [11,12,13]. Significant decreases in physical activity, increases in sedentary behavior, and disruptions in schedules and sleep quality in children and adolescents have been reported by several studies during the pandemic [13,14].

Several countries around the world have implemented measures such as large-scale vaccination and intervention to respond to the COVID-19 pandemic [1,15]. Approval has been granted for various SARS-CoV-2 vaccines [16,17]. The WHO’s target product profile for COVID-19 vaccines outlines that the vaccines in development are anticipated to be used for the active immunization of individuals of all ages in areas with ongoing outbreaks. These vaccines are expected to be combined with other control measures to prevent and mitigate the impact of COVID-19, aiming to reduce or terminate the outbreak [18,19]. Childhood vaccination protects young children and adolescents from disease and protects adults [20,21]. As of 2 November 2021, 23 vaccines have been listed under emergency use authorization in several countries; some have received full approval, and four are under the WHO emergency use listing [22,23]. One such vaccine is the protein subunit COVID-19 vaccine, IndoVac^®^, which has been developed by PT Bio Farma, Indonesia, in collaboration with Baylor College, Texas, USA [24,25]. The results of phase I, II, and III clinical trials in adults revealed safe and good immunogenicity [24,25]. Thus, this research is being conducted as a continuation of the booster studies in adults, where favorable immunogenicity was shown by the vaccine based on seropositive and seroconversion rates 14 and 28 days after the booster dose [26]. A significant increment in immunogenicity against the Omicron variant was exhibited by IndoVac^®^ at 3 and 6 months, based on the prebooster baseline [26]. IndoVac^®^ was tolerated well until the 6-month follow-up after the booster dose, and all reported adverse effects (AEs) were resolved [26].

Including children in vaccine trials would be beneficial for achieving herd immunity. Vaccinating populations aged ≥12 years will not only protect those vaccinated but also the unvaccinated individuals who are vulnerable to the disease. On 28 September 2022, IndoVac^®^ received EUA for individuals aged ≥18 years and obtained its full registration on 23 December 2023 from the Indonesia Regulatory Authority. This study mainly aimed to evaluate the immune response and safety to the SARS-CoV-2 neutralizing antibody of IndoVac^®^ before and 14 days after booster dose in adolescents aged 12–17 years.

## 2. Materials and Methods

### 2.1. Study Design

This study was a continuation of a booster study from adults [26]. This study used an open-label design because no vaccines were authorized for emergency use as a booster dose in adolescents aged 12–17 years in Indonesia. Participant eligibility was established before investigational product treatment commenced. After being informed about the study and signing the informed consent form (parent/legal guardian) and assent form (participants), the participants were evaluated through a medical history and physical exam. The investigator team checked the inclusion and exclusion criteria. For each participant, an inclusion number was allocated from 001 to 150. The participants were administered a booster dose of IndoVac^®^. For each dose administration, 0.5 mL of vaccine was injected intramuscularly into the left deltoid region by inserting the needle with a brisk dart-like action at a right angle to the skin surface. The participants remained under supervision at the site for at least 30 min after dose administration for evaluation of immediate reaction following each dose injection. For primary evaluation criteria, this research measured antibody titers for participants at baseline (pre-vaccination) and 14 days after the booster vaccination. Additionally, for secondary evaluation criteria, blood samples were collected at 3, 6, and 12 months after the booster vaccination. This study is registered with ClinicalTrials.gov (no. NCT05727215) and the Indonesia registry web portal (registry no. INA-M05FOH8). The inclusion criteria were clinically healthy children aged 12–17 years who have previously received a complete primary series of inactivated (Sinovac^®^) COVID-19 vaccine, with the last dose administered a minimum of 6 months but not longer than 18 months prior to inclusion. The parent/legal guardian and participant were informed properly regarding the study, signed the informed consent form (parent/legal guardian) and assent form (participant), and committed to comply with the instructions of the investigator and trial schedule. Conversely, the exclusion criteria were participants who concomitantly enrolled or scheduled to be enrolled in another trial; have received a booster dose of the COVID-19 vaccine; have a history of COVID-19 in the last 3 months (based on anamnesis or other examinations); have evolving mild, moderate, or severe illness, especially infectious disease or fever (body temperature ≥ 37.5 °C, measured with infrared thermometer/thermal gun); have a history of uncontrolled asthma; have allergies to vaccines or vaccine ingredients; have severe adverse reactions to vaccines, such as urticaria, dyspnea, and angioneurotic edema; have a history of uncontrolled coagulopathy or blood disorders contraindicating intramuscular injection; and have serious chronic diseases (e.g., serious cardiovascular diseases, uncontrolled hypertension and diabetes, liver and kidney diseases, malignant tumors), which, according to the investigator, may interfere with the assessment of the trial objectives.

### 2.2. Vaccines

The receptor-binding domain (RBD) vaccine was derived from the sequence of the Wuhan strain. The produced vaccine was IndoVac^®^ (protein subunit recombinant). Each 0.5 mL dose of vaccine contains 25 µg SARS-CoV-2 RBD subunit recombinant protein, 750 µg aluminum as an adjuvant, 750 µg CpG 1018 as an adjuvant, 2.226 mg NaCl, and 0.923 mg tris(hydroxymethyl) aminomethane. SARS-CoV-2 subunit recombinant protein vaccine contains SARS-CoV-2 receptor-binding domain (RBD) as an antigen. The clone of RBD protein was developed by the Texas Children’s Hospital Center for Vaccine Development at Baylor College of Medicine, USA. The protein was developed based on the sequence of the wild-type SARS-CoV-2 RBD amino acid, representing residues 331–549 of the spike protein (GenBank: QHD43416.1) of the Wuhan-Hu-1 isolate (GenBank: MN908947.3) [27,28,29]. The Omicron RBD binds to ACE2 with a 2.4-fold higher affinity at physiological conditions compared to the Wuhan strain, contributing to Omicron’s higher transmissibility and faster infection establishment [27].

### 2.3. Sample Size and Study Analysis

The sample calculation for this trial was based on the following assumptions related to a sample size calculation for before/after study using immunogenicity data. The minimum clinical difference to detect is two-fold the geometric mean titer (GMT) of anti-sRBD IgG and/or neutralizing antibody for pre- and post-test intervention at 0.30103 on a log scale (base 10) [30,31,32,33]. The standard deviation of the GMT of anti-sRBD IgG and/or neutralizing antibodies for pre- and post-test intervention on a log scale (base 10) is 1.0. Zα = Standard normal deviation for α = 1.9600. Zβ = Standard normal deviation for β = 1.2816. B = (Zα + Zβ)2 = 10.5074. C = (E/SΔ)2 = 0.090619. N = B/C = 116. Based on the assumption above, this trial should recruit 116 participants in each study arm to achieve 90% power at a two-sided 5% significance level. Anticipating 20% of dropout participants, we expanded to approximately 150 participants.

### 2.4. Immunogenicity Measurements

The result of immunogenicity was analyzed with 95% CI. For the neutralizing antibody, seropositive is defined as titer ≥ 4 dilutions or ≥46.03 IU/mL and seroconversion as a four-fold increase from baseline to 14 days after the booster dose or a change from seronegative to seropositive [26,28]. The conversion of neutralizing antibody titer to international unit (IU) was based on the equivalence of the dilution and Karber value from the method established and optimized by Bio Farma to the standard value of anti-SARS-CoV-2 immunoglobulin NIBSC code 21/234 (1:128 dilution = 128.00 Karber value = 1473.00 IU/mL). The optimization and validation results in Bio Farma showed that the standard titer is 1:4 dilution ≥ 46.03 IU/mL.

The IgG antibody titers were evaluated using chemiluminescent microparticle immunoassay for IgG antibody (CMIA) and microneutralization assay for neutralizing antibody. Seroconversion is evaluated following immunization with IndoVac^®^. The CMIA was performed by Prodia Laboratory, the first and only College of American Pathologists-accredited laboratory in Indonesia, which is also certified with NGSP and a year-long member of Clinical and Laboratory Standard Institutes with ISO 9001 [34] and ISO 15189 [35,36]. Neutralization assay was performed by the National Institute of Health Research and Development, Indonesian Ministry of Health (Balitbangkes Kemenkes), and Clinical Trial Laboratory of PT Bio Farma, which has been given laboratory accreditation by the WHO Regional South East Asia and Immunization and Vaccine Development with ISO 45001 [37] from PT Lloyd’s Register Indonesia, for and on behalf of Lloyd’s Register Quality Assurance Limited, United Kingdom [38].

The neutralization assay was conducted against the SARS-CoV-2 variant of concern (Omicron strain). A virus neutralization assay was developed in-house, in which we adopted the titration method from the WHO Polio Laboratory Assay. Virus neutralization using the Omicron strain is defined as the reduction in Omicron’s viral infectivity-mediated antibodies. The interpretation of endpoint neutralization titers was based on CPE (Cytopathic effects) by an inverted microscope. Sera from vaccinated volunteers were heat treated for 30 min at 56 °C. Four-fold serially diluted sera, from 1:4 to 1:512, was incubated with Omicron strain for 1 h at 37 °C. The mixture was subsequently incubated with Vero cells for 6 days. The interpretation of endpoint neutralization titers based on CPE (Cytopathic effects) by inverted microscope [39].

### 2.5. Safety Measurements

For safety assessments, the investigator evaluated the intensity (mild, moderate, or severe), duration, and the relationship of each adverse event to the trial vaccines. All the safety data were recorded in diary cards by the participants under the supervision of their parents. The safety data set includes the solicited and unsolicited adverse events that occurred from the beginning up to 28 days after the booster dose and serious adverse events that occurred from the beginning up to 12 months after the booster dose. Local and systemic reactions, whether expected or unexpected, that occurred within 30 min and on days 7, 14, and 28 after each immunization were assessed by interviewing participants during post-surveillance visits and documented in the diary card. Serious adverse events were monitored and evaluated throughout the study, up until 12 months after the final vaccination. Particularly, the body temperature was measured for 7 days after vaccination, in the evening and/or at the time of the febrile peak, and the highest temperatures were recorded in the diary card, expressed in Celsius degrees, using a thermometer. The trial team recorded the information in the electronic case report form (CRF).

### 2.6. Demographics and Baseline Characteristics

Overall, 151 participants were enrolled in the study from 28 February to 7 March 2023, and 150 participants received the IndoVac^®^ booster dose (Figure 1). Treatment compliance was defined as receiving a booster dose of IndoVac^®^ within the specified time windows.

Among 151 participants, 2 dropped out of the study, and 150 participants received a booster dose of IndoVac^®^ (Figure 2). Overall, there were 65 male participants (43.3%) and 85 female participants (56.7%), with a mean age of 14.71 ± 1.8 years. The majority of the participants were in senior high school (42.0%) (Table 1).

## 3. Results

### 3.1. Immunogenicity

#### 3.1.1. Primary Criteria

Administration of the booster dose of IndoVac^®^ to participants who previously received primary doses of CoronaVac^®^ increased neutralizing antibodies based on the prebooster baseline to the Omicron variant spike protein. The study showed a favorable increase in GMTs after a single booster dose of IndoVac^®^. The neutralizing antibody evaluation result against the Omicron strain was as follows (Figure 2). The neutralizing antibody GMTs (IU/mL) at baseline and 14 days after booster dose were 303.26 and 2661.21, respectively. The geometric mean fold rise (GMFR) was 8.77 (7.015–10.958). The seropositive rates of neutralizing antibodies at baseline and 14 days after booster dose were 92.67% and 100.00%, respectively. The seroconversion rate of neutralizing antibodies based on a four-fold increase at 14 days after booster dose was 69.78%, whereas based on the change from seronegative to seropositive, the rate was 100%.

#### 3.1.2. Secondary Criteria

##### Neutralizing Antibody

The neutralizing antibody GMT as the secondary endpoint is available as follows (Figure 2). The neutralizing antibody GMTs (IU/mL) at 3, 6, and 12 months after booster dose were 2021.09, 1172.74, and 868.73, respectively. The GMFR at 3, 6, and 12 months after the booster dose were 6.67 (5.217–8.536), 3.87 (3.068–4.886), and 2.87 (2.232–3.685), respectively. The seropositive rates of neutralizing antibodies at 3, 6, and 12 months after booster dose were all 100%. The seroconversion rates of neutralizing antibodies based on a four-fold increase at 3, 6, and 12 months after the booster dose were 58.70%, 48.55%, and 39.13%, respectively, whereas based on the change from seronegative to seropositive, the rates were all 100.00%.

##### IgG Antibody

The result of immunogenicity was presented with a 95% CI. The GMT and GMFR were presented as rates with a 95% CI analyzed using a *t*-test. For the anti-RBD IgG antibody, seropositive is defined as titer ≥ 50 AU/mL (≥7.1 binding antibody units per mL (BAU/mL)) and seroconversion as a change from seronegative to seropositive or a four-fold increase in anti-RBD antibody IgG titer compared to baseline if seropositive. The IgG antibody as the secondary endpoint is available as follows (Figure 3). The IgG antibody GMTs (BAU/mL) at baseline and at 14 days and 3, 6, and 12 months after the booster dose were 277.75, 3479.22, 2631.52, 1686.36, and 1243.33, respectively. Thus, GMFR at 14 days and 3, 6, and 12 months after booster dose were 12.53 (95% CI: 10.521–14.914), 9.49 (7.921–11.369), 6.08 (5.042–7.334), and 4.48 (3.71–5.18), respectively. The seropositive rates of IgG antibody at 14 days and 3, 6, and 12 months after booster dose were all 100%. The seroconversion rates of IgG antibody based on a four-fold increase at 14 days and 3, 6, and 12 months after the booster dose were 84.00%, 77.18%, 61.07%, and 51.68%, respectively (Figure 3).

#### 3.1.3. Comparison with Previous Study

This section evaluated the immunogenicity of IndoVac as a heterologous booster dose in adolescents compared with adults. Both age groups received complete primary doses of Sinovac for at least 6 months before enrollment [26].

Baseline neutralizing antibody titers were comparable between adults and adolescents. After the booster dose, geometric mean titers were 1266.69 (95% CI, 1014.731–1581.21) and 2661.21 (95% CI, 2275.086–3112.873) in adults and adolescents, respectively. Until 6 months after the booster dose, titers declined gradually but were still higher than baseline in both adults (GMT, 758.54; 95% CI, 637.309–902.827) and adolescents (GMT, 1172.74; 95% CI, 1009.258–1362.695) (Table 2).

Baseline IgG antibody was comparable between adults and adolescents. After the booster dose, geometric mean titers were 2817.08 (95% CI, 2460.41–3225.458) and 3479.22 (95% CI, 3182.65–3803.417) in adults and adolescents, respectively. Until 6 months after the booster dose, titers declined gradually but were still higher than baseline in both adults (GMT, 1090.63; 95% CI, 954.267–1246.474) and adolescents (GMT, 1686.36; 95% CI, 1504.862–1889.743) (Table 3).

### 3.2. Safety

In this study, the participants were vaccinated with a booster dose of IndoVac^®^. All participants were included in the safety population. Overall, the incidence rate of AEs until 28 days after booster dose was 82.7%, with a higher number of local events reported. Most reported solicited AEs were local pain (57.3%), followed by myalgia (40.0%), with mostly mild intensity (75.3%). Unsolicited AEs varied with each of the incidence rates < 10%, with mostly mild intensity (14.0%) (Figure 4). No adverse events of special interest (AESI) were noted. Until the 12-month follow-up after the booster dose, four SAEs occurred in the study. All the cases were assessed by the Data Safety Monitoring Board; the cases were identified as not related to the investigational products.

## 4. Discussion

This study revealed the immunogenicity and safety of a heterologous booster dose with a protein subunit recombinant COVID-19 vaccine (IndoVac^®^) in 150 participants aged 12–17 years. The COVID-19 vaccine may be required in children for several reasons. Most cases of COVID-19 are typically asymptomatic or mild in severity, the actual burden of the disease may be undermined, and vaccine trials initially were not prioritized in this population [40,41,42,43,44,45]. Moreover, with the emergence of the Delta and Omicron variants, infections in children appeared to become more frequent and severe [46,47]. Recent data from 2022 revealed that the proportion of infected children rose to 12.9% of reported global cases during a surge of the Omicron variant [46,48]. Thus, the COVID-19 booster vaccine is warranted in children and adolescents for protection against severe disease outcomes in children, especially those who have already been infected by the virus [49,50,51]. This study was the continuation of the previous study about IndoVac^®^ as a booster in adults that showed favorable immunogenicity and seroconversion rates of the vaccine 14 and 28 days after the booster dose, with the safety result being well-tolerated until the 6-month follow-up after the booster dose, and all reported AEs resolved [26,52,53,54].

This study revealed a favorable increase in GMTs after a single booster dose of IndoVac^®^, indicating non-inferiority immunogenicity compared to Covovax^®^ when assessed for neutralizing antibody titers against the Omicron variant. The seroconversion rates of neutralizing antibodies based on a four-fold increase at 14 days after the second dose for the IndoVac^®^ group was higher compared to the Covovax^®^ group. This is similar to the previous study about the heterologous booster vaccination of ZF2001 that showed that the GMTs of neutralizing antibodies against the prototype SARS-CoV-2 were increased from baseline [54]. This is similar to another study that showed that the post-booster vaccination GMTs increased from the baseline levels at days 14 and 28 [55]. Furthermore, this is consistent with the study by Liao Y et al. that reported that the third dose of the ZF2001 vaccine rapidly induced a significantly high degree of humoral immunogenicity and showed an increase in GMTs after 14 days of booster vaccination [56].

Moreover, this study showed that the IgG antibody GMT (BAU/mL) was increased on day 14 from baseline. The seropositive rates of IgG antibody at 14 days after booster dose was 100%. This is similar to the previous study that revealed that the neutralizing GMTs increased 14 days after the booster dose [57]. The second point of the secondary objective is to evaluate antibody persistence at 3, 6, and 12 months after the booster dose of IndoVac^®^. This study exhibited decreased IgG antibody GMTs (BAU/mL) at 3, 6, and 12 months after booster. The GMFR at 3, 6, and 12 months following the booster dose decreased over time. This result was similar to a study booster dose against the Omicron variant with different platforms in adolescents. The IgG results were decreased compared to 14 days after booster [58].

The seropositive rates of the IgG antibody at 3, 6, and 12 months after the booster were all 100%. This is consistent with a previous study that reported that 4–8 months after the two-dose vaccination, participants’ neutralizing GMTs remained detectable but decreased compared to that at 14 days post-vaccination shot [55]. However, further research about the COVID-19 vaccine booster in any age category is required to improve vaccine coverage [58]. The GMFR until 12 months after the booster was higher than that at baseline. This is similar to the previous study that showed that the GM titers of anti-spike protein antibodies were 10 times higher after the second dose of BNT162b2 with 6- to 13-week intervals [52]. However, this is in contrast with another study that showed GMT fold decreases after the 6-month follow-up of a booster dose of CoronaVac [59].

The phase III study of IndoVac^®^ conducted in Indonesia in 12- to 17-year-olds evaluated immunogenicity and safety. In line with the safety evaluation result, the results of the immunobridging evaluation support the use of protein subunit vaccines as safe and more tolerable options with a robust immune response [52]. This is similar to a previous study that demonstrated that the recombinant protein vaccine ZF2001 heterologous booster had a high immunogenicity and good safety profile in children and adolescents [60].

A previous study about IndoVac^®^ revealed that neutralizing antibody levels can be correlated and used to predict vaccine efficacy [26,51,52,61]. The overall efficacy of Covovax^®^ after the second dose was >80% in circulating variants at the time of the trials. That study showed that the neutralizing antibody levels of two doses of IndoVac^®^ can be immunologically associated with Covovax^®^’s efficacy [26,52]. Based on intracellular cytokines evaluation, no significant differences were observed between the IndoVac^®^ and Covovax^®^ groups post-vaccination [26,52]. Increases in IL-2-secreting CD8+ T cells and IL-2- and IL-4-secreting CD4+ T cells in the IndoVac^®^ and Covovax^®^ groups were observed 14 days after the second dose as compared to baseline (before vaccination) [26,51]. A study on the recombinant spike protein vaccine NVX-CoV2373 found that the levels of neutralizing antibodies and anti-S-binding IgG antibodies on day 35 (14 days after the second vaccine dose) are associated with the vaccine’s efficacy for NVX-CoV2373 [62]. Another study demonstrated statistically significant dose-dependent elevation of proliferating antigen-specific CD4+ T-helper cells and CD8+ T-killer cells in groups treated with 1/10 and 1/5 vaccine doses, respectively [63].

Both the neutralizing antibody and IgG antibody were markedly higher in the adolescents than in the adults at every timepoint [26]. Considering these factors, IndoVac is expected to provide a high degree of protection in adolescents, at least as much as in adults. This is similar to the previous study about the SARS-CoV-2 recombinant spike protein vaccine (SII-NVX-CoV2373) in children and adolescents. It is stated that the immune responses in children and adolescents were much higher than what has been observed in adults [63].

The last point of the secondary objective is to investigate the safety profile after a booster dose of IndoVac^®^. The present study showed that at the 12-month follow-up after the second dose, IndoVac^®^ was safe for use. This aligned with findings from another study, which indicated that heterologous boosting with the recombinant COVID-19 vaccine (Sf9 cells) was highly effective in protecting against symptomatic COVID-19 caused by different SARS-CoV-2 variants without presenting any significant safety issues [62]. Additionally, this finding was comparable to the study by Huang T et al., which found that administering a single dose of the ZF2001 heterologous booster after completing a two-dose series of inactivated vaccines in children and adolescents aged 3–17 years resulted in favorable safety and immunogenicity [54]. Another study also demonstrated that the ZF2001 recombinant protein vaccine, used as a heterologous booster, achieved high immunogenicity and maintained a favorable safety profile in this age group [60].

No safety concerns related to vaccination were found. During the study period, AESI was not noted. This is similar to the previous study that showed that one dose of the heterologous booster vaccination of ZF2001 after priming vaccination of two-dose inactivated vaccines in children and adolescents aged 3–17 years had good safety and immunogenicity [54]. Additionally, this is consistent with another study that revealed that the recombinant protein vaccine ZF2001 heterologous booster had high immunogenicity and a good safety profile in children and adolescents [60].

Limitations: The study had some limitations. This is the first study of the COVID-19 booster vaccine. Based on Indonesian regulations, only IndoVac^®^ can be used for boosters in Indonesia. Thus, we could not compare the other vaccines in this study. Effectiveness studies should be conducted to provide real-world data to describe the true efficacy of the vaccine. This study presented data of children aged 12–17 years only. Further research is warranted on the other groups of children aged <12 years.

## 5. Conclusions

The IndoVac^®^ vaccine has a favorable immunogenicity and safety profile. The immunogenicity-based neutralizing antibody and IgG antibody results showed that the vaccine was immunogenic. Furthermore, this study showed that the antibody titer decreases over time. The administration of IndoVac^®^ until 12 months after the booster dose was well-tolerated, and all AEs were reported as improved. These findings showed that the vaccine provides a safe and effective alternative for booster immunization, which potentially accelerates COVID-19 vaccination globally. Further research is required to determine whether another booster dose should be administered when antibody levels are decreased.

## Figures and Tables

**Figure 1 vaccines-12-00938-f001:**
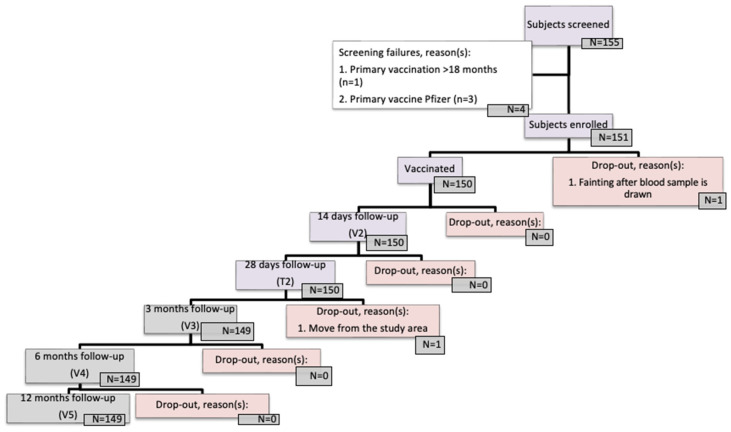
Participant disposition.

**Figure 2 vaccines-12-00938-f002:**
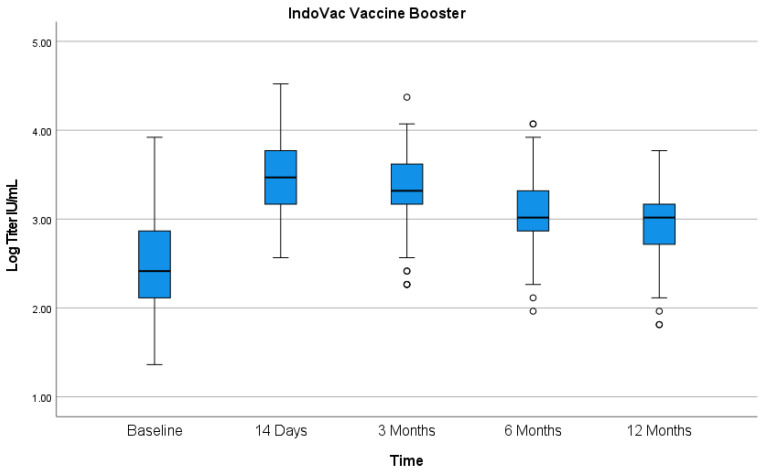
Neutralizing antibody titer before booster dose and at 14 days and 3, 6, and 12 months after booster dose.

**Figure 3 vaccines-12-00938-f003:**
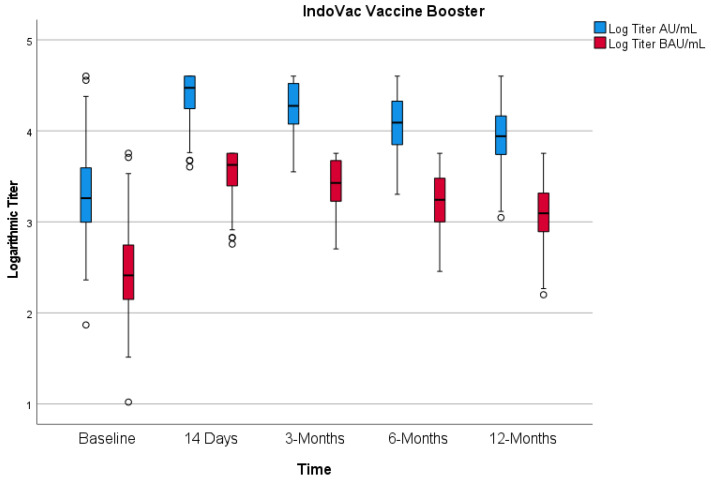
IgG antibody titer before vaccination and 14 days and 3, 6, and 12 months after the booster dose.

**Figure 4 vaccines-12-00938-f004:**
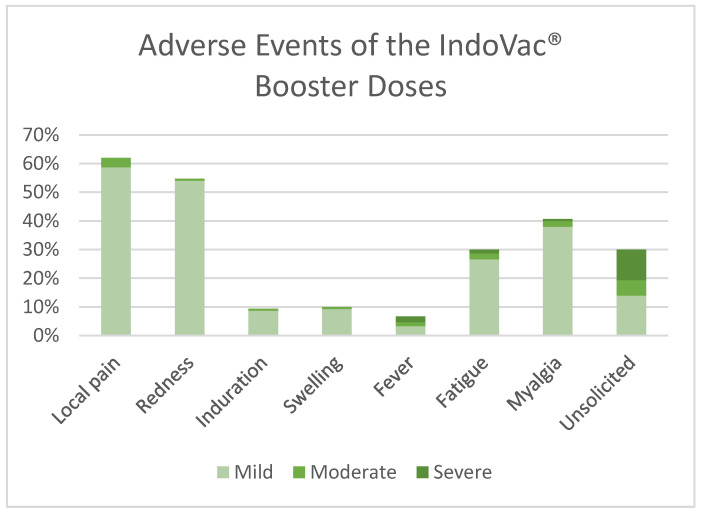
Adverse events of the booster dose of IndoVac^®^.

**Table 1 vaccines-12-00938-t001:** Demographic characteristics of the study.

Parameter	Total (N = 150)	
Mean age [years] (SD)	14.7	(1.8)
Mean height [cm] (SD)	155.6	(9.2)
Mean weight [kg] (SD)	50.3	(12.3)
BMI (kg/m^2^)	20.7	(4.4)
Sex, n (%)		
Male	65	(43.3%)
Female	85	(56.7%)
Education, n (%)		
Primary school	37	(24.7%)
Junior high school	50	(33.3%)
Senior high school	63	(42.0%)

Abbreviations: N, number of participants; SD, standard deviation.

**Table 2 vaccines-12-00938-t002:** Comparison of neutralizing antibody evaluation through 6 months after the booster dose in adults and adolescents.

Timepoint	Parameter	Booster Dose in Adults	Booster Dose in Adolescents
Before Vaccination	Seropositive Rate		
	n	113	139
	(%)	76.87	92.67
	(95% CI)	69.207–83.419	87.257–96.282
	GMT (IU/mL)	147.52	303.26
	(95% CI)	117.84–184.665	240.835–381.859
	Median	130.15	260.42
14 Days After Booster Dose	Seropositive Rate		
	n	145	150
	(%)	98.64	100
	(95% CI)	95.171–99.834	97.57–100
	GMT (IU/mL)	1266.69	2661.21
	(95% CI)	1014.731–1581.21	2275.086–3112.873
	Median	1041.57	2946
3 Months After Booster Dose	Seropositive Rate		
	n	144	149
	(%)	99.31	100
	(95% CI)	96.217–99.982	97.554–100
	GMT (IU/mL)	717.36	2021.09
	(95% CI)	594.713–865.309	1735.892–2353.143
	Median	736.5	2083.14
6 Months After Booster Dose	Seropositive Rate		
	n	140	149
	(%)	99.29	100
	(95% CI)	96.111–99.982	97.554–100
	GMT (IU/mL)	758.54	1172.74
	(95% CI)	637.309–902.827	1009.258–1362.695
	Median	736.5	1041.57

**Table 3 vaccines-12-00938-t003:** Comparison of IgG antibody evaluation through 6 months after the booster dose in adults and adolescents.

Timepoint	Parameter	Booster Dose in Adults	Booster Dose in Adolescents
Before Vaccination	Seropositive Rate		
	n	145	150
	(%)	98.64	100
	(95% CI)	95.171–99.834	97.57–100
	IgG (BAU/mL)		
	GMT	266.2	277.75
	(95% CI)	218.103–324.904	235.62–327.4
	Median	274.8	259.06
14 Days After Booster Dose	Seropositive Rate		
	n	147	150
	(%)	100	100
	(95% CI)	97.521–100.00	97.57–100
	IgG (BAU/mL)		
	GMT	2817.08	3479.22
	(95% CI)	2460.41–3225.458	3182.65–3803.417
	Median	3941.41	4225.86
3 Months After Booster Dose	Seropositive Rate		
	n	145	149
	(%)	100	100
	(95% CI)	97.488–100	97.554–100
	IgG (BAU/mL)		
	GMT	1804.82	2631.52
	(95% CI)	1590.084–2048.555	2378.491–2911.46
	Median	2017.64	2677.17
6 Months After Booster Dose	Seropositive Rate		
	n	141	149
	(%)	100	100
	(95% CI)	97.417–100	97.554–100
	IgG (BAU/mL)		
	GMT	1090.63	1686.36
	(95% CI)	954.267–1246.474	1504.862–1889.743
	Median	1052.9	1751.73

## Data Availability

Data will be available on the main site of the study. Contact the authors for future access.

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
