# Peer review of "Immunogenicity and Safety of SARS-CoV-2 Protein Subunit Recombinant Vaccine (IndoVac^®^) as a Heterologous Booster Dose against COVID-19 in Indonesian Adolescents"

_vaccines, 2024, doi:10.3390/vaccines12080938_

Round 1

Reviewer 1 Report

Comments and Suggestions for Authors

The manuscript by Fadlyana and colleagues evaluates the immunogenicity and safety of a subunit vaccine that consists of the peptide spanning the receptor binding domain (RBD). Overall, the manuscript was well-written, particularly the inclusion/exclusion criteria,  and highlights that a peptide of the Spike protein consisting of the RBD can boost antibody response against SARS-CoV-2.  Most of my comments are minor but I believe that statistical analysis would make for a stronger  manuscript

Comments:

Comment 1: Line 26: Please define AEs as adverse events.

Comment 2: line 73: At the end of this sentence add “titers.”

Comment 3: Line 74: Please delete “adverse events” and add “that” after (AEs).

Comment 4: Line 85: The word  “Adults” should be “adults.”

Comment 5: Page 3, Under 2.2 Vaccines: The RBD vaccine was derived from the sequence of the Wuhan strain.  It would interest the readers to know the amino acid differences between the RBD of the Wuhan and Omicron strains.

Comment 6: Lines 158-159: The authors conduct virus neutralization assays using the Omicron strain. The authors need to provide more details of the virus neutralization assay.

Comment 7: Figures 2, 3, and 4: The authors should analyze the data in these figures to determine the statistical significance.

Comment 8: Line 219: Please define “BAU.”

Comment 9: Line 235: Please define “ITT.”

Comment 10: Line 249: Replace “Most of the cases are..” with “Most cases of COVID-19 are…”

Comment 11: line 261: Replace “favorable increase of GMT” with “ a favorable increase of the GMT.”

Comment 12: In the sentence “This is similar to the other study that showed that the GMTs postbooster vaccination increased from the baseline levels by days 14 and 28” would read better as “This is similar to another study that showed that the post-booster vaccination GMTs increased from the baseline levels at days 14 and 28”

Comments on the Quality of English Language

No comments; was well written.

Author Response

Dear Reviewer 1,

Thank you for your input. We attach the details below.

Regards

Dr Eddy

Reviewer 2 Report

Comments and Suggestions for Authors

The paper reports on a trial that shows the good immunogenicity of boosters with Indovac highlighted by the presence of statistically-defined effective neutralisation titres in 100% of subjects at 12 months. The data are well-presented and important for post-pandemic vaccine development.

The antibody titration methods are not well described. The now published methods of the papers of Rusmil, et al and Nurdin et al. should be cited in the methods to show they have been used. The meaning of and the rationale for using the >46.03 IU/ml should be explained. 

The micro-neutralisation test needs to be defined giving the cell line, the readout and timing and interpretation. The previous papers did not define this and it should have been (referring to the polio manual is not sufficient).

The adverse event events scoring system is different to that used for the previous papers. Define what code 1, 2 and 3 mean. 

The finding that 10% of the subjects reported unsolicited severe unsolicited adverse events seems very important. This should be better reported and discussed.

Author Response

Dear Reviewer, 

Thank you for your input, We have been update it below on attachment.

Regards 

Dr Eddy

Reviewer 3 Report

Comments and Suggestions for Authors

Authors reported immunogenicity, titer of neutralizing antibody against an Omicron variant of SARS-CoV-2, and adverse effects in adolescents aged 12-17 years after receiving a booster of protein vaccine (IndoVac®) following initial (Sinovac®) COVID-19 vaccination. This study is currently ongoing and registered in ClinicalTrials.gov (no. 100 NCT05727215). The design and procedure are sound. However, the current manuscript did not compare other vaccine(s), and thus the efficacy or usefulness of this new protein vaccine has not been demonstrated.

Others issues:

1. The “Demographics and Baseline Characteristics” in the Results section is part of the Study Design” and should be moved to the Materials and Methods section.

2. Beside IgG, other serological and/or cellular responses also need to be measured. IgM, IgA, T cells, and/or cytokines also play important roles in short-term or long-term defense and adverse effects of vaccination.

3. It is not clear which Omicron subvariant(s) was/were tested for neutralizing.

4. The “adolescent” and “children” are different age groups. The two terms seem to be used interchangeably in the Results and Discussion sections to describe findings.  

5. Many abbreviations, such as SAE and GMT, do not have corresponding full names.

Comments on the Quality of English Language

Minor fix for gramma and English expression is needed.

Author Response

Dear Reviewer, 

Thank you for your input, below we attach the response.

Regards

Dr Eddy

Round 2

Reviewer 3 Report

Comments and Suggestions for Authors

The authors have not yet satisfactorily addressed this Reviewer's previous concerns including "the current manuscript did not compare other vaccine(s), and thus the efficacy or usefulness of this new protein vaccine has not been demonstrated", "The “Demographics and Baseline Characteristics” in the Results section is part of the Study Design”, and " Beside IgG, other serological and/or cellular responses also need to be measured. IgM, IgA, T cells, and/or cytokines also play important roles in short-term or long-term defense and adverse effects of vaccination". This study appears to be at early stage of vaccine development, and it is premature to report without data to show that the new vaccine can be a useful substitution or alternative for other well-established vaccines.

Comments on the Quality of English Language

Minor editorial correction is needed.

Author Response

Response to Reviewer 3 Comments Round 2

1. Summary

Authors reported immunogenicity, titer of neutralizing antibody against an Omicron variant of SARS-CoV-2, and adverse effects in adolescents aged 12-17 years after receiving a booster of protein vaccine (IndoVac®) following initial (Sinovac®) COVID-19 vaccination. This study is currently ongoing and registered in ClinicalTrials.gov (no. 100 NCT05727215). The design and procedure are sound. However, the current manuscript did not compare other vaccine(s), and thus the efficacy or usefulness of this new protein vaccine has not been demonstrated.

2. Questions for General Evaluation

Reviewer’s Evaluation

Response and Revisions

Does the introduction provide sufficient background and include all relevant references?

Can be improved

Thank you for reviewing our manuscript. We have been improved our article based on your comments.

Are all the cited references relevant to the research?

Must be improved

Is the research design appropriate?

Must be improved

Are the methods adequately described?

Must be improved

Are the results clearly presented?

Must be improved

Are the conclusions supported by the results?

Must be improved

3. Point-by-point response to Comments and Suggestions for Authors

Comments 1: The authors have not yet satisfactorily addressed this Reviewer's previous concerns including "the current manuscript did not compare other vaccine(s), and thus the efficacy or usefulness of this new protein vaccine has not been demonstrated"

.

Response 1: Thank you for pointing this out. At the time the clinical trial was conducted, there was no other vaccine registered from the NRA for booster doses in children aged 12–17. So, this research had an open-label design without any control vaccines. The efficacy of Indovac is available on previous research https://www.sciencedirect.com/science/article/pii/S0264410X24003992 and https://pubmed.ncbi.nlm.nih.gov/38675753/

Comments 2: Demographics and Baseline Characteristics” in the Results section is part of the Study Design

Response 2: Thank you for pointing this out. We agree to move it to study design

Comment 3: " Beside IgG, other serological and/or cellular responses also need to be measured. IgM, IgA, T cells, and/or cytokines also play important roles in short-term or long-term defense and adverse effects of vaccination".

Response 3 : Thank you for pointing this out. In this study, we evaluated IgG and MNT. T-cells and cytokines were evaluated in phase III in adults on previous research https://www.sciencedirect.com/science/article/pii/S0264410X24003992 and https://pubmed.ncbi.nlm.nih.gov/38675753/

Comment 4: This study appears to be at early stage of vaccine development, and it is premature to report without data to show that the new vaccine can be a useful substitution or alternative for other well-established vaccines

Response 4 : Thank you for pointing this out. Previous studies as primary vaccines, phase III in adults involving 4050 subjects and phase III in children involving 1050 subjects (not yet published) clinicaltrials.gov/study/NCT05546502, were already done. This vaccine has gone through clinical trials phases 1, 2, and 3 in adults and has been registered for 18 years and older.

4. Response to Comments on the Quality of English Language

Point 1: Minor editorial correction is needed..

Response 1: Thank you.

5. Additional clarifications

-

Round 3

Reviewer 3 Report

Comments and Suggestions for Authors

The Results are still insignificant to the field. Also, Figure 2 apparently duplicates the histograms of Baseline and 14-Days in Figure 3.  The authors indicate that the current study is a continuation of a study in adults of 18 years old or higher, which has been published and cited as #26 in the References. It would be significant if the current neutralizing antibody and IgG titers, and adverse effects in the adolescents of 12-17 years old are compared to the adults receiving primary CoronaVac® (also called SinoVac®) vaccine followed by the IndoVac® or BNT162b2 booster. It appears that the neutralizing antibody and IgG titers are higher in the adolescents, compared with adults receiving either the IndoVac® or BNT162b2 booster. The comparison is a new result and can be presented as figures and/or tables. Please also spell out the abbreviation when it appears first time in the Abstract or main body, and keep the consistency of abbreviation throughout the manuscript.  

Comments on the Quality of English Language

Few corrections as commented.

Author Response

Dear Reviewer, 

We have attached the answer. Thanks.

Regards

Dr Eddy

Round 4

Reviewer 3 Report

Comments and Suggestions for Authors

I spotted some typographical errors. Please run spelling check and change "(Figure 5)" in line 296 to "(Figure 4}". Also, add a sentence to the Abstract for the new comparison results.

Comments on the Quality of English Language

Minor revision is needed.

Author Response

Dear Reviewer, 

Thank you for your input, we attach the document below.

Regards

Dr Eddy
